

# Assessment tools for attention deficits in patients with stroke: a scoping review across components and recovery phases

Katsuya Sakai[1,2], Takayuki Miyauchi[3] and Junpei Tanabe[2,4]

[1] Department of Physical Therapy, Faculty of Health Sciences, Tokyo Metropolitan University, Tokyo, Japan
[2] Graduate School of Human Health Sciences, Tokyo Metropolitan University, Tokyo, Japan
[3] Faculty of Medical Sciences, Shonan University of Medical Sciences, Kanagawa, Japan
[4] Department of Physical Therapy, Hiroshima Cosmopolitan University, Hiroshima, Japan

## ABSTRACT

**Background:** Attention deficits are common in patients with stroke, making the assessment of attention functions crucial for improvement. A previous review reported on attention deficit assessments using specific components in patients with stroke. However, this study only included randomized controlled trials (RCTs) and did not encompass the attention assessments included in the observational study. Therefore, we reviewed and categorized the assessments used for attention deficits in patients with stroke according to specific attention components including RCTs and observational studies.

**Method:** In this study, we adhered to the scoping review guidelines. The population, concept, and context of this study were stroke; attention deficits, RCTs, observational studies, and assessments; and components (focused, selective, sustained, spatial, divided, visual, and auditory attention) and phase (acute, subacute, and chronic), respectively. Two reviewers independently screened articles at the title, abstract, and full-text levels based on inclusion and exclusion criteria using four databases and the Rayyan software. Furthermore, we identified the study design, sample size, duration since stroke onset, and assessment tools were identified.

**Results:** Out of 1,423 articles, we selected 35. The study designs included observational studies (80%) and RCTs (20%) and a total of 2,987 patients. The age range was 40.0 ± 7.7 to 83.6 ± 9.7 years. Twenty-four assessment tools were identified, mainly including the Trail Making Test Part A, Test of Everyday Attention, and other assessments (40%, 11.4%, and 62.8%, respectively). Regarding the five components of attention, there were 10 assessments were used each for sustained and selective attention (28.6%), and six each for alertness and divided attention (17.1%). Spatial attention was assessed using only one tool (2.9%).

**Conclusions:** We identified various assessment tools for analyzing attention deficit in patients with stroke and mapped them by component. This scoping review would be useful for selecting assessment methods for patients with stroke with attention deficits.

Corresponding author
Katsuya Sakai, k.sakai@tmu.ac.jp

## INTRODUCTION

Attention is a complex and multidimensional function involving sustained attention, selection, and other functions associated with arousal and task demands. Therefore, it has been explained using several models (*Sohlberg & Mateer, 2001*; *Loetscher et al., 2019*). The most common is the neuropsychological model proposed by *Sohlberg & Mateer (2001)*. The model is divided into five categories based on characteristics of attentional function. First, focused attention is based on attention function, followed by sustained, selective, and alternating attention, and finally, the most advanced and complex, divided attention. *Loetscher et al. (2019)* replaced the alternating attention of the five attention classifications of *Sohlberg & Mateer (2001)* with spatial attention using the following five components: focused, sustained, selective, spatial, and divided attention (Table 1). This added spatial attention to general attention, incorporates the multiple aspects of the attention function. In addition, attention includes visual and auditory aspects (*Giard et al., 2000*; *Souto & Kerzel, 2021*). By proposing these models, it became possible to evaluate attentional functions by component, to study their relationship with physical functions, and to calculate the prevalence of each component.

Attention deficits in stroke were associated with motor and balance function, falls, limited independence in activities of daily living (ADL), and reduced quality of life (*Barker-Collo et al., 2010b*; *Hyndman & Ashburn, 2003*; *Hyndman, Pickering & Ashburn, 2008*). Furthermore, *Hyndman, Pickering & Ashburn (2008)* reported that attention deficits correlated with hospital discharge rates at 12 months post-stroke. As mentioned above, post-stroke attention deficits affected several patients, with 46–92% of patients affected in the acute phase (*Stapleton, Ashburn & Stack, 2001*) and 24–51% affected at the time of hospital discharge (*Hyndman, Pickering & Ashburn, 2008*). As mentioned earlier, attention has a total of seven aspects, each of which needs to be considered to determine its prevalence and relationship to physical function in patients with stroke.

Prevalence rates vary by attention component in patients with stroke. *Spaccavento et al. (2019)* used the Test of Attention Performance (TAP) to investigate the alertness and selective attention of patients with stroke. They discovered that 5.6% and 44.4% of patients had impaired focused attention and selective attention, respectively. *Hyndman, Pickering & Ashburn (2008)* investigated the prevalence of attention components in patients with stroke using the Test of Everyday Attention (TEA). They reported that 51%, 37%, 36%, and 37% patients with stroke had divided, sustained, auditory, and visual attention deficits, respectively. In addition, *Loetscher et al. (2019)* reviewed the assessment of attention deficits in patients with stroke and the effects of cognitive rehabilitation for each component of attention. Consequently, they clarified the assessment and effects for four components, excluding spatial attention. TAP and simple reaction time were used to assess arousal, while TAP and Trail Making Test (TMT) part A assessed selective attention. The Integrated Visual and Auditory Continuous Performance Test (IVT-CPT) and Full scale Attention Quotient assessed sustained attention, and TAP and Paced Auditory Serial Addition Test (PASAT) assessed distributive attention. However, the study by *Loetscher et al. (2019)* only included randomized controlled trials (RCTs), and did not include

**Table 1 Components of attention and definition.**

| Attention | Definition | Functional example |
|---|---|---|
| Focused attention | Ability and readiness to respond | Response to warning signals |
| Sustained attention | Ability to focus on specific stimuli while ignoring irrelevant stimuli | Reading while people talk in the back-ground |
| Selective attention | Ability to maintain attention over a prolonged period of time | Driving a car for long distances |
| Spatial attention | Ability to detect and deploy attention to all sides of space | Attending to people sitting on left and right side of the table |
| Divided attention | Ability to multitask and to divide attention between 2 or more tasks | Talking on the telephone while cooking |

observational studies. There are many observational studies in the previous studies (*Spaccavento et al., 2019*; *Barker-Collo et al., 2010a*; *Hyndman & Ashburn, 2003*; *Hyndman, Pickering & Ashburn, 2008*), and by including observational studies, it would be possible to chart a more useful assessment by component. Furthermore, charting the assessment of the components of attention at each stage (acute, subacute, chronic) of stroke will allow the characteristics of the assessment to be captured, and this will be important information for diagnosis and rehabilitation. However, this has not been shown in previous research. In this scoping review, we categorized the assessments used for post-stroke attention deficits by component and phase in both observational studies and RCTs, to aid in selecting assessments for attention deficits and for verifying the effects of rehabilitation interventions.

## MATERIALS AND METHODS

In this study, we followed the Preferred Reporting Guidelines for Systematic Reviews and Meta-Analyses (PRISMA) statement 2018 for scoping reviews (*Tricco et al., 2018*). We utilized the scoping review methodology of *Arksey & O'Malley (2005)*, *Levac, Colquhoun & O'Brien (2010)*, and the Joanna Briggs Institute (https://jbi.global/scoping-review-network/resources). The study protocol adhered to the Open Science Framework (osf.io/y6a35). This study was conducted in five stages following the studies conducted by *Arksey & O'Malley (2005)* and *Levac, Colquhoun & O'Brien (2010)*. They include (1) identifying the research question, (2) identifying relevant studies, (3) selecting the studies, (4) charting the data, and (5) collating, summarizing, and reporting the results.

We used the population, concept, and context (PCC) framework in this study (Table 2). Four databases (PubMed, Web of Science, CINAHL, and Ovid) were used to search for articles published from their inception to May 23, 2024. Search terms are listed in File S1.

The inclusion criteria were as follows: (1) the presence of stroke; (2) age > 18 years; (3) studies written in English; (4) RCTs or observational studies; and (5) assessments regarding attention function. The exclusion criteria included the following: (1) the absence of stroke; (2) other high brain dysfunctions (aphasia, unilateral spatial neglect (USN), apraxia, attention-deficit/hyperactivity disorder); (3) visual deficits; (4) age < 18 years; (5) studies not written in English; (6) nonhuman participants; (7) editorials, commentaries, conference abstracts, letters, reviews, book, case reports, and feasibility studies; (8) undescribed assessments.

**Table 2 The population, concept, and context framework.**

| | |
|---|---|
| P (population) | Stroke |
| C (concept) | Attention deficits, randomized controlled trials, observational studies |
| C (context) | Assessments, <br> Components (focused, selective, sustained, spatial, divided, visual, and auditory attention), <br> Phase (acute, subacute, and chronic) |

Two reviewers (Katsuya Sakai and Takayuki Miyauchi) independently reviewed the articles during the primary and secondary screening using the inclusion and exclusion criteria with the Rayyan software (https://www.rayyan.ai/). Rayyan integrates articles, removes duplicate articles, and distinguishes between accepted and rejected articles. In addition, Rayyan can mask or share the results of other reviewers. The third reviewer (Junpei Tanabe) independently accepted or rejected the article based on the inclusion and exclusion criteria of the study during disagreements between the first (Katsuya Sakai) and the second reviewer (Takayuki Miyauchi). The two reviewers conducted the charting independently after the secondary screening, after charting the three articles. Subsequently, we held a meeting to ensure that there were no differences in the extracted items or content and that any areas of disagreement in the charting were corrected. Furthermore, all articles were charted, and any areas of disagreement were corrected during the meetings.

The components of attention were categorized according to the five components identified by *Loetscher et al. (2019)*. Visual and auditory attention were categorized using previous studies (*Giard et al., 2000*; *Souto & Kerzel, 2021*). Assignment of attention by construct was conducted using a previous study (*Strauss, Sherman & Spreen, 2006*).

The classification of the phase of stroke was defined as follows: acute phase within 30 days, subacute phase from 30 days to less than 6 months, and chronic phase after 6 months, according to a previous study (*Duncan et al., 1992*). This is because these classifications reflect the stages of stroke recovery.

# RESULTS

Figure 1 presents the PRISMA diagram of the review process. A total of 1,423 articles were extracted from the four databases, excluding 129 duplicates. Furthermore, 1,294 articles were selected for primary screening. Sixty-six studies were subjected to secondary screening, wherein 31 articles were excluded for reasons such as non-English language, other diseases, lack of attention assessments, or inappropriate study design. Ultimately, 35 articles were deemed eligible for mapping (Table 3) (*Barker-Collo et al., 2009*; *Hyndman, Pickering & Ashburn, 2008*; *Hasanzadeh Pashang et al., 2021*; *Pearce et al., 2016*; *Navarro et al., 2020*; *Barker-Collo et al., 2010a*; *Rasquin, Welter & van Heugten, 2013*; *Murakami et al., 2014*; *Tanikaga et al., 2018*; *Duffin et al., 2012*; *Spaccavento et al., 2019*; *D'Imperio et al., 2021*; *Hyndman & Ashburn, 2003*; *Chen et al., 2013*; *Hochstenbach et al., 1998*; *Jaywant et al., 2018*; *Zucchella et al., 2014*; *Peers et al., 2020*; *Kim et al., 2011*; *Arikawa et al., 2023*; *Fishman, Ashbaugh & Swartz, 2021*; *Pinter et al., 2019*; *Ten Brink et al., 2024*; *Kim, Ko & Woo, 2013*; *Graber et al., 2019*; *Maeneja et al., 2023*; *Schaapsmeerders et al., 2013*;

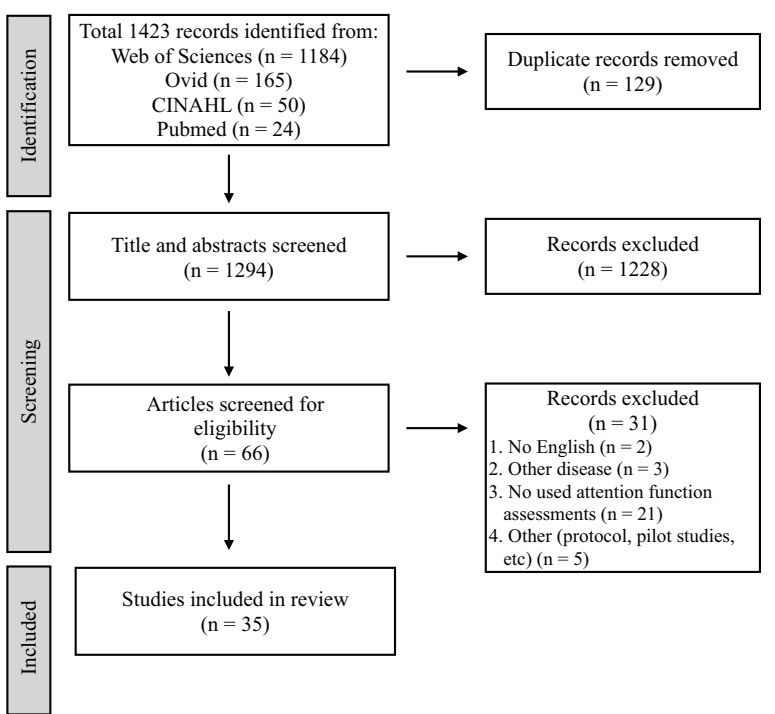

**Figure 1 Preferred reporting items for systematic reviews and meta-analyses.**

*Chen et al., 2009*; *Facchini et al., 2023*; *Lord et al., 2006*; *Yorozuya et al., 2022*; *Stebbins et al., 2008*; *Van Zandvoort et al., 2005*; *Lin et al., 2018*; *Verhoeven et al., 2011*).

## Study design, sample size, age, and phase of stroke

The study designs included observational studies 80% (28 out of 35 articles) and RCTs 20% (seven out of 35 articles). The total sample size was 2,987 participants, ranging 12–606 participants per study. The age range was 40.0 ± 7.7 to 83.6 ± 9.7 years. Regarding the stroke phase, 14.3% (five out of 35 articles), 22.9% (8 out of 35 articles), and 40% (14 out of 35 articles) included the acute phase (<1 month), the subacute and chronic phase (>6 months), respectively, with time frames ranging from 8.3 days to 8.5 years. Of the other eight studies, seven and one studies did not state the time after onset and were in the subacute to chronic phase, respectively.

## Assessments

Studies were charted using 24 assessment tools (Table 3), including the TMT Part A (40%, 14 out of 35 articles), TEA (11.4%, four out of 35 articles), Integrated Visual Auditory (IVA)-CPT, d2 Test of Attention, PASAT, TAP (5.7%, two out of 35 articles), and others (62.9%, 22 out of 35 articles).

## Components of attention functions

Table 4 and Fig. 2 show the attention assessments by component and exhibit the percentages of the assessment components, respectively.

**Table 3 Overview of all articles.**

| Authors | Year | Design | Sample size | Age | Time since stroke | Assessment tools | Five Components based on Loestcher et al. | Except five components |
|---|---|---|---|---|---|---|---|---|
| Barker-Collo et al. (2009) | 2009 | RCT | 78 | 67.7 (15.6)–70.2 (15.6) | 18.48 (11.95)–18.58 (7.62) months | IVT-CPT | – | Auditory, Visual attention |
| Hyndman, Pickering & Ashburn (2008) | 2008 | Observational | 122 | 70.2 (12.5) | – | TEA | Sustained attention, Selective attention, Divided attention | – |
| Hasanzadeh Pashang et al. (2021) | 2021 | Observational | 20 | 53.9 (9.7)–57.7 (12.2) | 11.90–20.30 months | IVA + Plus | – | Auditory, Visual attention |
| Pearce et al. (2016) | 2016 | Observational | 22 | 68.2 (12.2) | 262.65 (92.07) days | Psychomotor vigilance task | Sustained attention | – |
| Navarro et al. (2020) | 2020 | RCT | 43 | 51.7 (18.1)–52.9 (10.6) | 374.3 (229.9)–433.6 (258.5) months | Conner's CPT, d2 Test of Attention, Color trail test, Digit span, Spatial span | Focused attention (Conner's CPT, d2 Test of Attention, Color Trail Test Part A), Sustained attention (Conner's CPT, d2 Test of Attention), Selective attention (d2 Test of Attention, Color Trail Test Part A, Digit span, Spatial Span), Divided attention (Color Trail Test Part B) | – |
| Barker-Collo et al. (2010a) | 2010a | Observational | 94 | 68.2 (15.7) | 17.99 (10.05) days | IVT-CPT, TMT Part A, PASAT | Selective attention (TMT Part A) | Auditory (IVT-CPT, PASAT), Visual attention (IVT-CPT) |
| Rasquin, Welter & van Heugten (2013) | 2013 | Observational | 42 | 57.1 (7.7) | – | TMT Part A | Selective attention | – |
| Murakami et al. (2014) | 2014 | Observational | 115 | 66.4 (9.89) | – | CAT | Focused attention, Sustained attention, Selective attention, Divided attention, Spatial attention | Auditory, Visual attention |
| Tanikaga et al. (2018) | 2018 | Observational | 24 | 74.9 (8.7)–83.6 (9.7) | 53.5 (21.0) days–6.4 (9 years) | TMT Part A, PASAT | Selective attention (TMT Part A) | Auditory (PASAT) |
| Duffin et al. (2012) | 2012 | Observational | 16 | 65.4 | 5–60 days | TMT Part A | Selective attention | – |
| Spaccavento et al. (2019) | 2019 | Observational | 204 | 62.8 (10. 6) | 122.7 (93.1) days | TAP | Focused attention, Selective attention, Divided attention | – |
| D'Imperio et al. (2021) | 2021 | Observational | 29 | 62.4 (11.9) | 7.18 (4.60) months | The attentional matrices test | Selective attention | – |
| Hyndman & Ashburn (2003) | 2003 | Observational | 48 | 68.4 (11.7) | 71.5 (35–88) months | TEA | Sustained attention, Selective attention, Divided attention | – |
| Chen et al. (2013) | 2013 | Observational | 90 | 57.5 (14.2) | 30.0 (25.2) months | TEA | Sustained attention, Selective attention, Divided attention | – |
| Hochstenbach et al. (1998) | 1998 | Observational | 12 | 53.4 | 49.4 (5–104) days | TMT Part A, The WAIS digit symbol subtest | Focused attention (The WAIS digit symbol subtest), Selective attention (TMT Part A), Sustained attention (The WAIS digit symbol subtest) | – |

 

| Authors | Year | Design | Sample size | Age | Time since stroke | Assessment tools | Five Components based on Loestcher et al. | Except five components |
|---|---|---|---|---|---|---|---|---|
| *Jaywant et al. (2018)* | 2018 | Observational | 95 | 67.9 (14.9) | 8.3 (9.8) days | TMT Part A | Selective attention | – |
| *Zucchella et al. (2014)* | 2014 | Observational | 92 | 64–70 | – | TMT Part A, Attentive matrices | Sustained attention (Attentive Matrices), Selective attention (Attentive Matrices, TMT Part A) | – |
| *Peers et al. (2020)* | 2020 | Observational | 23 | 59.0 (10.6) | 8.5 (4.7) years | Partial and Whole Report TVA paradigm | – | Visual attention |
| *Kim et al. (2011)* | 2011 | RCT | 30 | 70.7 (6.6)–71.4 (5.2) | 130.7 (121.4)–148.3 (188.2) days | TMT Part A, Computerized Neuropsychological Test (visual continuous performance test) | Selective attention (TMT Part A) | Visual attention (Computerized Neuropsychological Test) |
| *Arikawa et al. (2023)* | 2023 | Observational | 17 | 63.4 (9.4) | 31.1 (49.4) months | TMT Part A, PASAT | Selective attention (TMT Part A) | Auditory (PASAT) |
| *Fishman, Ashbaugh & Swartz (2021)* | 2021 | RCT | 72 | 66.4 (12.3)–70.4 (11.2) | At least 3 months | TMT Part A | Selective attention | – |
| *Pinter et al. (2019)* | 2019 | Observational | 114 | 44.5 (9.5) | 10 (8–11) days | TMT Part A | Selective attention | – |
| *Ten Brink et al. (2024)* | 2024 | Observational | 262 | 58.5 (11.4) | 35.18 (78.89) days | Hierarchical visual processing, TMT Part A | Selective attention (TMT Part A), Divided attention (Hierarchical visual processing) | |
| *Kim, Ko & Woo (2013)* | 2013 | RCT | 38 | 52.4 (2.6)–58.9 (3.1) | Chronic | TMT Part A | Selective attention | – |
| *Graber et al. (2019)* | 2019 | Observational | 70 | 63.3 (14.7)–70.3 (13.1) | 6 months | TAP | Focused attention, Selective attention, Divided attention | – |
| *Maeneja et al. (2023)* | 2023 | RCT | 34 | 55.1 (6.6)–57.0 (10.2) | Subacute to chronic | d2 Test of attention | Focused attention, Sustained attention, Selective attention | – |
| *Schaapsmeerders et al. (2013)* | 2013 | Observational | 606 | 40.0 (7.7)–40.1 (8.0) | – | Verbal series attention test | – | Auditory |
| *Chen et al. (2009)* | 2009 | Observational | 39 | 58.9 | 712.6 (14–2,626) days | CCPT, Digit vigilance test | Sustained attention | – |
| *Facchini et al. (2023)* | 2023 | Observational | 133 | 59.6 (13.4)–65.4 (13.9) | – | TMT Part A | Selective attention | – |
| *Lord et al. (2006)* | 2006 | RCT | 27 | 61.0 (11.6) | 45.8 (34.2) months | TEA | Sustained attention, Selective attention, Divided attention | – |
| *Yorozuya et al. (2022)* | 2022 | Observational | 51 | 72.1 (11.0) | 3 months | VCT | Selective attention | – |
| *Stebbins et al. (2008)* | 2008 | Observational | 51 | 63.1 (8.2)–67.4 (9.7) | 3–6 months | Digit forward | – | – |
| *Van Zandvoort et al. (2005)* | 2005 | Observational | 73 | 56.0 (16.0) | 4–20 days | TMT Part A | Selective attention | – |

(Continued)

| Authors | Year | Design | Sample size | Age | Time since stroke | Assessment tools | Five Components based on Loestcher et al. | Except five components |
|---|---|---|---|---|---|---|---|---|
| *Lin et al. (2018)* | 2018 | Observational | 90 | 56.9 (12.4)–57.1 (12.6) | 2.3 (0.9–17.6) months | The computerized digit vigilance test | Sustained attention | – |
| *Verhoeven et al. (2011)* | 2011 | Observational | 111 | 63.7 (14.4) | – | TMT Part A | Selective attention | – |

**Note:**
RCT, Randomized controlled trial; TMT, Trail Making Test; IVT-CPT, Integrated Visual Auditory Continuous Performance Test; TEA, Test of Everyday Attention; IVA +Plus, Integrated Visual and Auditory Continuous Performance Test; CPT, Continuous Performance Test; PASAT, Paced Auditory Serial Addition Test; CAT, Clinical Assessment for Attention; TAP, Test of Attentional Performance; CCPT, Conners' Continuous Performance Test II; VCT, The visual cancelation task.

**Table 4 Component of attention function and assessments.**

| Component | Assessment tools |
|---|---|
| Focused attention | d2 Test of Attention, TAP, CAT, Color Trail Test Part A, Conner's CPT, The WAIS digit symbol subtest |
| Sustained attention | TEA, d2 Test of Attention, Attentive Matrices, CAT, CCPT, Conner's CPT, Digit Vigilance Test, Psychomotor vigilance task, The Computerized Digit Vigilance Test, The WAIS digit symbol subtest |
| Selective attention | TMT Part A, TEA, d2 Test of Attention, TAP, Attentive Matrices, CAT, Color Trail Test Part A, Digit span, Spatial Span, VCT |
| Divided attention | TEA, TAP, CAT, Color Trail Test Part B, Divided attention test, Hierarchical visual processing |
| Spatial attention | CAT |
| Auditory attention | IVA + Plus, IVA-CPT, PASAT, Verbal Series Attention Test, TEA |
| Visual attention | IVA + Plus, IVT-CPT, Partial and Whole Report TVA paradigm, Neuropsychological Test, TEA |

**Note:**
CAT, Clinical Assessment for Attention; CPT, Continuous Performance Test; CCPT, Conners' Continuous Performance Test II; TAP, Test of Attentional Performance; TEA, Test of Everyday Attention; TMT, Trail Making Test; VCT, Visual Cancelation Task; IVA-CPT, Integrated Visual Auditory Continuous Performance Test; IVA + Plus, Integrated Visual and Auditory Continuous Performance Test; PASAT, Paced Auditory Serial Addition Test.

Focused attention assessments included the following six tools: the d2 Test of Attention (5.7%), TAP (5.7%), Clinical Assessment for Attention (CAT, 2.9%), Color Trail Test Part A, Conner's Continuous Performance Test (CCPT, 2.9%), and Wechsler Adult Intelligence Scale (WAIS) Digit Symbol Subtest (2.9%).

Sustained attention assessments comprised 10 tools: TEA (11.4%), d2 Test of Attention (5.7%), Attentive Matrices (2.9%), CAT, CCPT (2.9%), Conner's CPT (2.9%), Digit Vigilance Test (DVT, 2.9%), Psychomotor Vigilance Task (2.9%), Computerized DVT (2.9%), and WAIS Digit Symbol Subtest (2.9%).

Selective attention assessments encompassed 10 tools: TMT Part A (40%), TEA (11.4%), d2 Test of Attention (5.7%), TAP (5.7%), Attentive Matrices (2.9%), CAT (2.9%), Color Trail Test Part A (2.9%), Digit Span (2.9%), Spatial Span (2.9%), and Visual Cancelation Task (VCT, 2.9%).

Divided attention assessments included six tools: TEA (11.4%), TAP (5.7%), CAT (2.9%), Color Trail Test Part B (2.9%), Divided Attention Test (2.9%), and Hierarchical Visual Processing (2.9%).

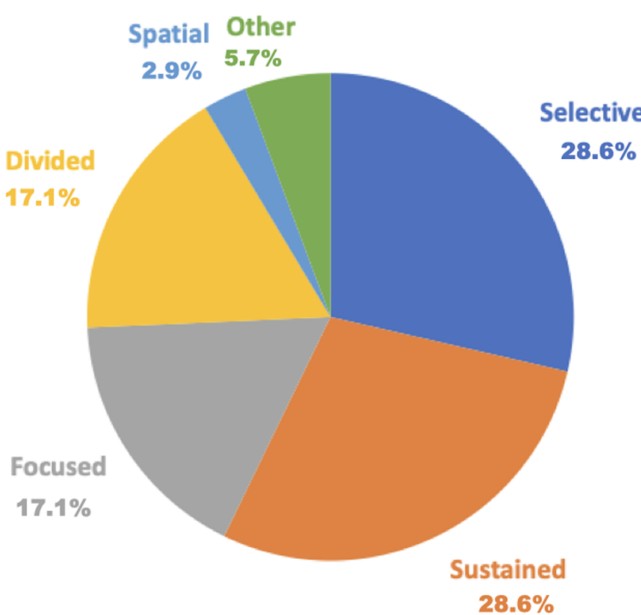

**Figure 2 Percentage of assessment by the five attention components.** Blue indicates selective attention at 28.6% (10 out of 35 articles); orange indicates sustained attention at 28.6% (10 out of 35 articles); gray indicates focused attention at 17.1% (six out of 35 articles); yellow indicates divided attention at 17.1% (six out of 35 articles); light blue indicates spatial attention at 2.9% (one out of 35 articles); and green indicates duplicate tools at 5.7% (two out of 35 articles).

| Table 5 Attention assessments and phases of stroke. | |
| --- | --- |
| **Phase of stroke** | **Assessment tools** |
| Acute (1< month) | TMT Part A, PASAT, IVA-CPT, The WAIS digit symbol subtest |
| Subacute (1> month to <6 month) | TMT Part A, PASAT, TAP, TEA, The WAIS digit symbol subtest, Computerized Neuropsychological Test, d2 Test of Attention, VCT, Digit forward, The Computerized Digit Vigilance Test, Hierarchical visual processing |
| Chronic (>6 months) | TMT Part A, PASAT, Psychomotor vigilance task, CPT, CCPT, Digit forward, Digit Vigilance Test, IVA-CPT, The attentional matrices test, TEA, d2 Test of Attention, Color Trail Test, IVA + Plus, Digit span, Spatial Span, Partial and Whole Report TVA paradigm, TAP |

Note:
CPT, Continuous Performance Test; CCPT, Conners' Continuous Performance Test II; TAP, Test of Attentional Performance; TEA, Test of Everyday Attention; TMT, Trail Making Test; VCT, Visual Cancelation Task; IVA-CPT, Integrated Visual Auditory Continuous Performance Test; IVA + Plus, Integrated Visual and Auditory Continuous Performance Test; PASAT, Paced Auditory Serial Addition Test.

Spatial attention assessment included only one tool which was, CAT (2.9%).

Auditory attention assessments comprised five tools (14.3%): IVA + Plus, IVA-CPT, PASAT, Verbal Series Attention Test, and TEA.

Visual attention assessments included four tools: IVT + Plus, IVT-CPT, Partial and Whole Report TVA Paradigm, and neuropsychological tests. The most used assessment for visual and auditory attention was the IVA-CPT and PASAT (5.7 and 8.6%, respectively).

## Stroke phase specific attention assessments

Table 5 exhibits the assessments according to the phase of stroke. The assessments used regardless of stage were TMT Part A and PASAT.

The acute phase (<1 month) included four tools: TMT Part A, PASAT, IVA-CPT, and the WAIS Digit Symbol Subtest.

The subacute phase comprised 11 tools: TMT Part A, PASAT, TAP, TEA, WAIS Digit Symbol Subtest, Computerized Neuropsychological Test, d2 Test of Attention, VCT, Digit Forward, Computerized DVT, and Hierarchical Visual Processing.

The chronic phase (>6 months) encompassed 18 tools: TMT Part A, PASAT, Psychomotor Vigilance Task, Conner's CPT, CCPT, Digit Forward, Digit Vigilance Test, IVA-CPT, Attentional Matrices Test, TEA, d2 Test of Attention, Color Trail Test, IVA + Plus, Digit Span, Spatial Span, Partial and Whole Report TVA Paradigm, TAP, and d2 Test of Attention.

## DISCUSSION

In this study, we aimed to chart the assessment of attention deficits in patients with stroke alongside the components of attention and stroke phases. Thirty-five articles from four databases were identified for this scoping review. The study design included 80% and 20% observational studies and RCTs, respectively. Twenty-four assessment tools were utilized; the TMT Part A (40%) and TEA (11.4%) were the most common. Other tools included the IVA-CPT, d2 Test of Attention, PASAT, and TAP (5.7%). The TMT Part A was the most frequently utilized assessment tool for attention function. Selective and sustained attention assessments were used in 28.6% of the studies. Numerous assessment tools were used in the chronic-phase studies (18 tools) of the stroke phase. We charted 34 articles that had not been identified in a previous study (*Loetscher et al., 2019*) by incorporating a cross-sectional study and identifying some research gaps. In this scoping review, we charted the assessments utilized for post-stroke attention deficits by component and phase, in both observational studies and RCTs, to aid in selecting assessments for attention deficits and verifying the effects of rehabilitation interventions.

### Assessments

In this study, we charted 24 types of attention function assessments, which exceeds those reported in prior studies. For example, *Loetscher et al. (2019)* reviewed only RCTs and identified 17 tools. Therefore, this study could chart additional studies that were missed in prior research by including both observational studies and RCTs. Among the 35 selected articles, TMT Part A was the most frequently utilized assessment tool, seen in 14 articles (40%). Furthermore, TMT Part A measures the motor speed and selective attention and is used in efficacy validation reviews of cognitive rehabilitation (*Loetscher et al., 2019*; *Tsiakiri et al., 2024*). TMT Part A is popular owing to its simplicity and short assessment time despite its limited scope in assessing functions. This may contribute to its frequent utilization in studies. Notably, TMT part A contains assessment factors other than those mentioned above (*Tsiakiri et al., 2024*). TEA was used in 11.4% of the articles, and it was not charted in previous studies (*Loetscher et al., 2019*; *Tsiakiri et al., 2024*). This is the first time TEA was charted in a study. TEA comprises eight tests that simulate ADL, each capable of assessing sustained, selective, and visual and auditory attention (*Chen et al., 2013*). Attention deficits in patients with strokes have been linked to difficulties in ADL

(*Hyndman & Ashburn, 2003*; *Hyndman, Pickering & Ashburn, 2008*). Therefore, assessing the ADL capabilities of patients with stroke with attention deficits is crucial. This necessity probably explains why TEA was the second most commonly used tool, following the TMT Part A. Consequently, the inclusion of cross-sectional studies and RCTs in this study have resulted in the presentation of new results in the review that were not included in previous reviews (*Loetscher et al., 2019*).

## Components

This scoping review charted the five attention components reported by *Loetscher et al. (2019)*, as well as visual and auditory attention, totaling seven components. CAT was extracted for the first time in this study, and it is the only tool that can comprehensively assess all seven components. CAT comprises eight different attention types, making it a versatile assessment tool (*Murakami et al., 2014*). Therefore, it may be advantageous to assess all components using a single tool. However, the sensitivity and specificity of each CAT item are unknown; therefore, it should be considered in conjunction with other assessments.

More assessment tools have been used to evaluating deficits in selective and sustained attention than in other components of attention. Sustained attention is the ability to maintain consistent task performance over a long period (*Sohlberg & Mateer, 1987*). Patients with stroke often exhibit deficits in sustained attention, which are associated with to declines in postural control, mobility, and ADL (*Hyndman & Ashburn, 2003*; *Hyndman, Pickering & Ashburn, 2008*; *Lin et al., 2018*; *Pearce et al., 2016*). In addition, deficits in sustained attention hinder motor function recovery (*Robertson et al., 1997*). Sustained attention is a fundamental aspect of attention; therefore, it is frequently assessed in studies considering daily life functions (*Langner & Eickhoff, 2013*). Selective attention disorders are prevalent and can significantly impact physical function (*Barker-Collo et al., 2009*; *McDowd et al., 2003*). Therefore, assessing both selective and sustained attention is crucial, as many assessment tools may be employed for these purposes.

Only the CAT was charted for spatial attention. Because this study excluded USN, the number of assessments charted was limited. The spatial attention test of CAT is a cancelling task, and this is a similar assessment to the Behavioural Inattention Test (BIT) (*Wilson, Cockburn & Halligan, 1987*). When USN is excluded, as in this study, the BIT is not charted, so it is clear that the BIT is used as an assessment specific to USN. However, the BIT is also highly useful as an assessment for attention disorders other than USN, such as extinction and drawing tests. Therefore, the assessment of spatial attention should be used not only for USN, but also for a wider range of disorders. By charting the attention function into seven components, we were able to provide a more detailed evaluation. This will lead to a more multifaceted approach to evaluating the attention function, and will be useful for diagnosis and rehabilitation.

## Stroke phase

The greatest number of tools were used during the chronic phase (18 tools) of the stroke phase. In addition, TMT Part A and PASAT were frequently used regardless of the stroke

phase. These results supported that of the previous review (*Loetscher et al., 2019*). PASAT, which involves adding numbers based on auditory information, is known for its ease of utilization (*Barker-Collo et al., 2009*) and PASAT is effective for evaluating divided attention (*Mathias & Wheaton, 2007*) and can be used in all stroke phases. The charting of common and different assessments at different stages is important for longitudinally assessing symptom changes and validating the effectiveness of rehabilitation.

The assessments used in the acute phase were TMT, PASAT, IVT-CPT, and the WAIS digit symbol subtest. These were assessments that focused (the WAIS digit symbol subtest), selective (TMT), auditory and visual attention (PASAT and IVT-CPT). All of these assessments were characterized by their short testing times and their ability to be administered quickly. We considered this a valid result. During the acute phase, the patient's condition is unstable, and natural recovery is rapid, making it easy for physical functions to change. For this reason, it may be easier to use simple and highly sensitive assessments such as these.

Eleven different assessments were used for the subacute phase. Unlike the acute phase, various attention function assessments were used to enable evaluation of all seven attention components. In addition, the TEA, which evaluates attention function in daily life (*Chen et al., 2013*), was included, and we assumed that assessments that take into account the patient's destination were being used.

Eighteen assessments were used in the chronic phase, and they covered all of the components of attention. The prevalence of assessments during the chronic phase may be a result of the stabilization of physical function recovery, enabling ease of detecting attention deficits in observational studies and excluding the impact of spontaneous recovery in intervention studies. The fact that many evaluations have been charted is thought to indicate that it is easy to incorporate any kind of attention evaluation in the chronic phase, making it possible to evaluate the patient's condition in more detail.

## Limitation of the study

This scoping review had some limitations. First, it focused exclusively on patients with stroke and, therefore, did not chart the assessment of other disease attention functions. However, given the high prevalence of attention deficits in patients with stroke, assessing these deficits is crucial for effective rehabilitation interventions. Second, this review included few spatial attention assessments and excluded patients with USN to distinguish USN from attention disorders, as USN is classified as directed attention. Previous studies did not include USN and reviewed it separately (*Loetscher et al., 2019*; *Umeonwuka, Roos & Ntsiea, 2022*). Thirdly, we did not perform a reliability test when charting. When investigating the agreement rate of the charted data, it would have been possible to create a highly reliable chart by performing a reliability test. In our next study, we will perform a reliability test. Fourthly, this study did not investigate the sensitivity, specificity, reliability, or validity of each assessment. This is because in order to create a chart of these, it was necessary to add terms associated with them. Adding terms would reduce the number of articles that took into account the components and phases, which is the purpose of this study. Therefore, in the next study, we will conduct a review focusing on sensitivity and

specificity based on the results of this study. Finally, a quality assessment of the articles was not conducted owing to the differing study designs, which prevented standardized findings. This is due to the difficulty generalizing the results following the differing study designs. Future studies should include a quality assessment, particularly focusing on observational studies. In addition, this study charted the assessment tools and components of attention deficits in patients during different stroke phases. Therefore, future research should explore rehabilitation methods for specific attention components and stroke phases.

## CONCLUSIONS

In conclusion, in this study, we charted attention function assessments, component-specific assessments, and assessments according to phases of stroke in patients with stroke. TMT Part A and TEA were frequently used, with various assessments conducted for different components and stroke phases. In the chronic phase, many assessments were used, and it was shown that it was possible to assess various types of attention, while in the acute phase, simple and highly sensitive assessments were charted. Considering the results of this review, expanding the assessment of spatial attention and conducting assessments of attention functions associated with ADL during the acute phase may lead to advances in predicting patient outcomes, diagnosis, and the development of rehabilitation strategies. This study serves as an important guide for selecting appropriate assessments for patients with stroke and attention deficits.

### Funding
The authors received no funding for this work.

### Competing Interests
The authors declare that they have no competing interests.

### Author Contributions
- Katsuya Sakai conceived and designed the experiments, performed the experiments, analyzed the data, prepared figures and/or tables, authored or reviewed drafts of the article, and approved the final draft.
- Takayuki Miyauchi analyzed the data, prepared figures and/or tables, authored or reviewed drafts of the article, and approved the final draft.
- Junpei Tanabe analyzed the data, authored or reviewed drafts of the article, and approved the final draft.

### Data Availability
    The raw measurements are available in the Supplemental File.

## Supplemental Information

Supplemental information for this article can be found online at http://dx.doi.org/10.7717/peerj.19163#supplemental-information.

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
