# Peer review of "Assessment tools for attention deficits in patients with stroke: a scoping review across components and recovery phases"

_PeerJ, doi:10.7717/peerj.19163_

## Round 0.1 · original submission · Major Revisions

Dear Authors,

Thank you for submitting your manuscript to PeerJ.
The Reviewers pointed out a number of methodological and structural limitations in the manuscript that should be addressed in order for the paper to be condiserated for publication. Please consider all comments and suggestions from Reviewers, in order to improve the quality of the manuscript.

Kind regards,
Marialaura Di Tella

Reviewer 1 ·

Basic reporting

I read with interest the manuscript entitled “Assessments tools mapping for attention deficit in patients with stroke: A scoping review”. In this paper, the authors aimed at summarising the existing literature on the methodologies to evaluate the attentional deficits in patients with stroke. In this article there are many issues that need to be addressed to improve the overall quality of the manuscript.
English quality: authors should carry out a global revision of the English language. Employed terms are sometimes ambiguous or inadequate, making it difficult for the reader to immediately grasp what the authors wanted to convey. The reviewer suggests having a colleague who is proficient in English and familiar with the subject matter review your manuscript or contact a professional editing service.
Introduction & background: The introduction is occasionally confused, redundant in the content and scarcely organised. Moreover, the aim of the scoping review appears poorly framed by the theoretical background, which should be revised to better reflect the scope of the study and the literature gap that it aims to address.
References: The literature is globally well-referenced and relevant.
The structure conforms to PeerJ standards.
The review is within the scope of the journal and the field lacks a recent review of available tools for attention assessment.

Experimental design

Methods: the description of employed methodology should be revised. Indeed, some information needs to be reported in a more structured and ordered manner (e.g., lines from 135 to 144 and from 155 to 160 could be summed up in tables and briefly described in text). Moreover, some paragraphs could be summarised (e.g., from lines 145 to 154).
Regarding the choice of the keywords, the authors may specify how they limited the semantic search as well as the reason why they used different keywords for different databases (attention disorders was not used in Pubmed for example).
Furthermore, more technical English terms should be employed to describe how the review process was carried out (e.g., the term “mapping” to express the data extraction phase is inadequate). Finally, an assessment of the quality of included studies is missing, which severely limits the usefulness of the work, leaving open the question as to which clinical instrument is most appropriate for the assessment of attention, in its components, in patients with stroke.
Results: overall, the results section is quite difficult to navigate. Results are outlined as a long list without a clear structure. Authors need to better organise and summarise the results section, preferring tables and figures over long unstructured lists.
They may describe the results related to assessment tools based on the attentional component that each instrument evaluates, to reduce redundancy.
Figure 2 may be improved by reporting the percentages in the figure.

Validity of the findings

The discussion session gives a global picture of the study’s findings, summarising what has emerged from the literature review. However, no clear benefit to existing literature is stated and results appear globally inconclusive. A more explicit and critical reflection on the contribution of the present work to existing literature is required for authors to complete the manuscript.
The authors identify limitations and weaknesses of the present study, remaining literature gaps and future directions.

Additional comments

Overall, the present manuscript would benefit from a global rewriting and reorganization, to improve the quality of the paper. Moreover, a substantial rephrasing due to a level of English below the standard of quality required for publication is needed. As is, the manuscript is not fit for publication.

Reviewer 2 ·

Basic reporting

no comment

Experimental design

no comment

Validity of the findings

no comment

Additional comments

Review for Assessments tools mapping for attention deficit in patients with stroke: A scoping review

1. Out of 1,423 articles, 35 were selected. The study designs included observational studies (80%) and RCTs (20%).
Would the combination of observational studies and RCTs lead to unreliable conclusions?
For observational studies, they can provide valuable information about associations and trends in the real world. However, they are often subject to confounding factors and biases, which may affect the reliability of the conclusions.
For RCTs, they are designed to minimize biases through randomization and control groups. They generally provide stronger evidence for causal relationships.

2. Is attention deficit common among stroke patients? Does it have an impact on the development of stroke, such as recurrence and death? What is the practical significance of the clinical application of multiple assessment tools?

3. This study identified various assessment tools for attention deficit in stroke patients and mapped them by component. Are these tools truly effective? Are there comparabilities among different methods?

Reviewer 3 ·

Basic reporting

1. some parts of the introduction can be improved in terms of writing style and the use of the English language.
2. Although there may be limited previous studies related to the topic, however, it would be interesting if authors can include more recent/up-to-date citations.
3. some terms can be improved in the tables and figure. For pie-chart, please include the numbers and percentage.
4. The review article highlights a very specific yet interesting area for stroke rehabilitation. I believe it aligns within the scope of the journal.
5. Review on specific area such posed by this article is relatively new. Although the authors only assess which tools frequently used, however, it would be more interesting to see which among those are efficient in detecting attention deficits in stroke patients.
6. introduction section can be improve, please elaborate more on the components instead of just giving the definitions. Emphasize why the components by Loetchscher was chosen as the guide

Experimental design

1. Article content is within the Aims and Scope of the journal and article type
2. the search methods were rigorous and clear.
3. method section could be improve in terms of write-up and arrangements.
4. sources are properly cited.

Validity of the findings

1. Conclusions are well stated, linked to original research question and identify the unresolved gap of knowledge.

Additional comments

Title
Suggestion to remove word “mapping” or change it to be more specific “"Assessment Tools for Attention Deficits in Stroke Patients: A Scoping Review Across Components and Recovery Phases"

Introduction
Line 83-85 : the reference use to describe the prevalence of attention deficit post-stroke are from 2001 and 2008 literatures. Please find a latest update on this topic.
Line 92-93: please cite Loetscher et al (2019). This part of introduction is very crucial in this study as it is the guide to review the articles collected. Please elaborate more on the components instead of just giving the definitions. Emphasize why the components by Loetchscher is chosen as the guide.
Line 92: Please include a pictograph/diagram to visualize the components of attention function.
Line 96, 96: Ability should not start with capitalizing the word.
Line 104-109 : clarify or rewrite this part as the write-up is confusing. Please improve the English language.
Line 116: does this statement refers to Loetscher study? “However, their study only included randomized…”
Line 121: Instead of ‘Mapping”, please use a more traditional or straightforward term, you could use alternatives like "categorizing," "organizing," or "overview."

Materials and method
Lien130-131: please check the format of citation for website.
Line 135- 138: Please arrange PCC one by one, as the current write -up arrangement is confusing. population is stroke patients. Please specify whether its primary stroke patients, secondary stroke patients or both.
Line 145-154: explain the function of Rayyan software. Please clarify the criteria of articles were remove in primary screening. The inclusion and exclusion criteria should be mention first before explaining the selection process.
Table 1: “outcome” should be “assessment tool”. “five components” should be “components of attention based on Loetscher et al. (2019)”. “except five components” should be “other components”
Table 2: “outcome” should be “assessment tool”. I would suggest to combine Table 2 and Table 3.
Table 4: “Phases” should be “Phase of Stroke”. “outcome” should be “assessment tool”. Include the duration for the stroke phases (acute stroke (<1 month))
Figure 2: please include numbers and percentage inside the pie chart.

Discussion
Line 241: “From four online literature databases..”
The discussion is well elaborated and covers all the data which has been analysed in the result section. Although there may be limited previous studies related to the topic, however, it is interesting if authors can include more recent/up to date citations as part of the discussion.

Conclusion
Conclusion is well summarized.

---

## Round 0.2 · Major Revisions

Dear Authors,

Thank you for submitting your manuscript to PeerJ.

Reviewer 4 pointed out a series of issues that should be addressed in order for the paper to be condiserated for publication. Please consider all comments and suggestions from Reviewers, in order to improve the quality of the manuscript.

Kind regards,
Marialaura Di Tella

Reviewer 2 ·

Basic reporting

The author has addressed the reviewer's questions and suggestions very well. The current version is suitable for acceptance and publication. Thank you.

Experimental design

The author has addressed the reviewer's questions and suggestions very well. The current version is suitable for acceptance and publication. Thank you.

Validity of the findings

The author has addressed the reviewer's questions and suggestions very well. The current version is suitable for acceptance and publication. Thank you.

Additional comments

The author has addressed the reviewer's questions and suggestions very well. The current version is suitable for acceptance and publication. Thank you.

Reviewer 4 ·

Basic reporting

The manuscript titled “Assessment Tools for Attention Deficits in Patients with Stroke: A Scoping Review Across Components and Recovery Phases” offers a valuable and relevant review of attention assessments following stroke. However, several areas would benefit from further clarification and refinement. Below are specific suggestions for improvement:
The manuscript provides a solid background on attention deficits post-stroke. However, the introduction is too condensed and should more clearly articulate the research gap and objectives. Additionally, some sections would benefit from more explicit connections between cited studies and the rationale for this review to improve transitions and enhance readability (the paragraphs jump between models of attention, stroke-related deficits, and prevalence rates).
I recommend strengthening the justification for this review by clearly distinguishing it from previous studies (e.g., Loetscher et al., 2019) and expanding on the theoretical framework beyond Sohlberg & Mateer (2001) (the rationale for including observational studies alongside RCTs should be more explicitly stated, and the need to categorize assessments by stroke phase should be introduced earlier and more clearly).

Experimental design

The manuscript appropriately follows PRISMA guidelines, and the search strategy is clearly defined, although the protocol has not been registered. To enhance replicability, please specify the date when the search was conducted in the Methods section. Additionally, the data extraction and charting process lacks a report on inter-rater reliability, and it is unclear how studies were categorized by stroke phase—particularly given the underrepresentation of certain phases (e.g., acute). I would suggest including a kappa statistic for inter-rater reliability and providing a rationale for the time-frame classification of stroke phases.

Validity of the findings

In the results and discussion sections, while the summary of 35 studies is informative, it lacks a critical analysis of how different assessment tools compare in terms of reliability, validity, and clinical applicability. Including an appraisal of the identified tools—considering their sensitivity, specificity, advantages, and limitations—would significantly strengthen the review. Additionally, the underrepresentation of some components (e.g., spatial attention assessment tools) should be examined in greater depth, as it could indicate a gap in research.
Overall, it seems that the Authors do not explicitly tie back to the objectives stated in the introduction (the review aims to categorize assessments by stroke phase, but it does not critically discuss whether certain assessments are more appropriate for specific phases). The discussion still primarily lists assessment tools and prevalence rates rather than providing an in-depth evaluation (for example, it mentions that more tools were used in the chronic phase, but it does not explain why this matters).
In the conclusion section, I recommend reinforcing the discussion on how these findings should influence clinical decision-making and future research. For example, it would be valuable to highlight key unanswered questions in the field. At present, the conclusions are somewhat generic, and the potential clinical impact of the findings of this scoping review is underdeveloped.

---

## Round 0.3 · accepted · Accept

Dear Authors,

Thank you for submitting your manuscript to PeerJ.
The Reviewers' comments and suggestions have been adequately addressed. Therefore, the paper can be accepted for publication.

Kind regards,
Marialaura Di Tella

Reviewer 4 ·

Basic reporting

I have no further comments for the Authors.

Experimental design

I have no further comments for the Authors.

Validity of the findings

I have no further comments for the Authors.